# Narcissistic Personality Traits and Sexual Satisfaction in Men: The Role of Sexual Self-Esteem

**Annalisa Anzani** *, **Marco Di Sarno**, **Rossella Di Pierro** and **Antonio Prunas**

Department of Psychology, University of Milano-Bicocca, 20126 Milan, Italy;
m.disarno@campus.unimib.it (M.D.S.); rossella.dipierro@unimib.it (R.D.P.); antonio.prunas@unimib.it (A.P.)
* Correspondence: annalisa.anzani@unimib.it

**Abstract:** Research on pathological narcissistic traits and sexuality are mainly focused on the dysfunctional aspects of sexuality. The present study aims to explore the relationship between narcissistic traits and sexual satisfaction in men, testing whether sexual self-esteem mediates this association. Participants included 212 men who completed measures of grandiose and vulnerable narcissistic traits, sexual self-esteem, and sexual satisfaction. Results highlight how vulnerable narcissistic traits are negatively associated with sexual satisfaction. This association is fully mediated by sexual self-esteem. On the contrary, grandiose narcissistic traits are not directly associated with sexual satisfaction, but with sexual self-esteem only, which explains the indirect effect of grandiose traits on sexual satisfaction. In conclusion, sexual self-esteem in personality configurations with high pathological narcissistic traits accounts for the relationship between narcissistic traits and sexual satisfaction.

**Keywords:** narcissistic personality traits; sexual satisfaction; vulnerable narcissism; grandiose narcissism; sexuality





## 1. Introduction

Sexual satisfaction is a key component of sexual health that contributes to the overall well-being of individuals [1]. In more detail, sexual satisfaction consists of the personal evaluation of one's sexual life, which is influenced by past sexual experiences, and present and future expectations [2]. Sexual satisfaction has been associated with romantic relationship satisfaction, both in cross-sectional and longitudinal studies [3–5]. Moreover, the quality of sexual life, including sexual satisfaction, has been shown to predict the overall quality of both present and future romantic relationships [6].

Personality traits are an important standpoint for understanding and predicting sexuality [6], including sexual satisfaction. Narcissism, in particular, has been linked to sexual life since its introduction in the field of psychology [7]. Historically, it was inherently described as a form of pathological sexuality, including excessive autoerotism [8] and a type of "libidinal" investment characterized by a relatively exclusive focus on the self [9].

Since then, the conceptualization of pathological narcissism has evolved substantially. The construct is nowadays conceptualized as a pathology of self-esteem regulation, organized around antagonistic themes [10,11], and defined by self-regulatory deficits and maladaptive strategies to cope with ego-threats [11–14]. Pathological narcissism is usually identified with aspects of grandiosity (i.e., inflated self-image), callousness, and lack of empathy [15–17], especially in its affective component [18]. However, it also refers to vulnerable themes, including emotional and behavioral dysregulation in response to self-enhancement failures (e.g., shame, low self-esteem), in the context of an antagonistic interpersonal style [10,13,19]. Traits of grandiose and vulnerable narcissism are distributed in the general population with varying levels [20,21], sometimes—but not necessarily—reaching the threshold for a full-blown narcissistic personality disorder [22].

Grandiose and vulnerable narcissism have different nomological correlates [23]. For instance, the former is usually related to high self-esteem, whereas the opposite is true for the latter [24]. Moreover, measures of grandiose narcissism are associated with higher global life satisfaction in nonclinical samples [25–27], whereas the opposite is usually the case for vulnerable narcissism [26]. In a way, high levels of grandiose narcissism are more related to distress "induced" in significant others, whereas high levels of vulnerable narcissism relate to subjective distress [10,28].

Contemporary conceptualizations of narcissism no longer flatten the construct on the sexual and libidinal domain. In spite of this, sexual life in narcissists remains a topic of interest for both researchers and clinicians. Indeed, antagonistic features in narcissistic personalities can have a significant impact on one's relational and sexual life. Aspects such as sexual exploitation and entitlement, low sexual empathy, and an inflated sense of sexual skills even define what has been described as "sexual narcissism", a set of components of narcissistic personality styles that are activated within specific sexual situations [29].

Both global and sexual narcissism are important for the study of sexual functioning and satisfaction. Traditionally, their role was explored in light of the negative consequences on sexual life [30,31], both for oneself and others. For instance, sexual narcissism was found to uniquely predict sexual aggression on the one hand and reduced sexual self-satisfaction and functioning on the other [6,29,31]. As for global narcissism, studies showed that both grandiose and vulnerable narcissism negatively affected relationship satisfaction in heterosexual couples [32]. Grandiose narcissism was related to increased socio-sexuality and low relational commitment in a number of studies [33,34], and resulted in extensive consumption of pornography [35] and sexual aggression [36]. Moreover, pathological narcissism (including both grandiose and vulnerable manifestations) was found to be associated with lower relationship satisfaction in both partners in heterosexual nonclinical couples [37], as well as reduced sexual satisfaction, intimacy, and orgasmic responsivity in male partners [30]. Interestingly, the association of narcissism with such sexual outcomes was mediated by a higher motive for affirmation in sex in men [30].

These findings notwithstanding, it has to be noted that both sexual and global narcissism were also associated with positive sexual outcomes in a few studies. For instance, McNulty and Widman [6] found that sexual skills were associated with higher sexual satisfaction in heterosexual nonclinical couples at the beginning of their marriage. In a similar sample, sexual skills were also positively related to sexual functioning and, in turn, to higher life satisfaction in women [31]. Moreover, grandiose narcissism was found to correlate with sexual satisfaction positively, and to reduce the magnitude of the association between sexual satisfaction and self-esteem [38], suggesting that narcissists' self-esteem is not contingent upon sexual satisfaction. Hence, when the grandiose and vulnerable manifestations of narcissism are investigated separately, they show differential patterns of associations with relevant outcomes in the sexual domain, with vulnerable narcissism being more unequivocally related to reduced self-satisfaction in sex.

The aim of this study is to investigate the relationship between pathological narcissistic traits and sexual satisfaction in the last six months, in a nonclinical sample of men. Indeed, meta-analytic evidence shows that men have higher levels of narcissism compared to women on average [39]. In doing so, this study takes into account the difference between grandiose and vulnerable narcissistic traits and, in line with recent studies on sexual satisfaction [38], investigates their specific effects separately. In addition to previous research, the current study investigates whether sexual self-esteem mediates the relationship between narcissism and sexual satisfaction. In fact, previous findings show that global self-esteem mediates the link between narcissism and global life satisfaction [27], such that those high in grandiose narcissism report higher self-esteem and, in turn, higher levels of life satisfaction. In this sense, the present study extends such findings to the sexual domain in order to identify a potential mechanism that underlies the association between narcissism and sexual satisfaction. Our study is somewhat exploratory in nature. However, based on the literature reviewed in this introduction, we expect vulnerable narcissism to be

related to lower satisfaction, with this association being mediated by low sexual self-esteem. We also expect grandiose narcissism to be positively associated with sexual satisfaction, with the mediation of high sexual self-esteem.

## 2. Methods

### 2.1. Participants and Procedure

Participants included 212 Italian men (see Table 1 for sample demographics), with an overall mean age of 30.4 (SD = 7.8; range = 18–54 years). There was limited sexual diversity within the sample, where 11.3% of men identified as a sexual minority and 88.7% as heterosexual. In addition to self-expressed sexual identity, we used a measure of sexual attraction and behavior in the last six months to reflect the complexity of defining sexual orientation (see Table 2) [40]. Most of the individuals in the sample stated they were in a couple (68.4%). The remaining declared themselves to be single (31.6%). Participants were recruited through flyers posted on various online resources and social media. After providing informed consent, participants were asked to complete an online survey seeking to explore "the quality of sex life and its relationship with specific personality characteristics". No incentive was offered for participation. The study was approved by the Ethics Committee of the University of Milano-Bicocca, with the ethical code number: 180, 2015.

**Table 1.** Socio-demographic characteristics of the sample.

| Demographics | *N* (%) |
| --- | --- |
| **Relationship Status** | |
| Single | 67 (31.6) |
| Cohabiting couple | 71 (33.5) |
| Non-cohabiting couple | 74 (34.9) |
| **Marital Status** | |
| Single | 166 (78.3) |
| Married | 35 (16.5) |
| Separated/divorced | 9 (4.3) |
| **Educational Level** | |
| Lower than high school | 9 (4.2) |
| High school degree | 103 (48.6) |
| University degree | 76 (35.8) |
| Post-graduate education | 24 (11.3) |
| **Working status** | |
| Employed | 137 (64.6) |
| Unemployed | 7 (3.3) |
| Student | 68 (32.1) |

**Table 2.** Dimensions of sexual orientation.

| Dimensions of Sexual Orientation | *N* (%) |
| --- | --- |
| **Sexual Identity** | |
| Heterosexual | 188 (88.7) |
| Bisexual | 7 (3.3) |
| Homosexual | 15 (7.1) |
| Other | 2 (0.9) |
| **Sexual Attraction** | |
| Exclusively to women | 182 (85.8) |
| Predominantly to women more than accidentally to men | 2 (0.9) |
| Predominantly to women only accidentally to men | 8 (3.8) |

**Table 2.** *Cont.*

| Dimensions of Sexual Orientation | N (%) |
|---|---|
| Equally to women and men | 1 (0.5) |
| Predominantly to men more than accidentally to women | 2 (0.9) |
| Predominantly to men only accidentally to women | 6 (2.8) |
| Exclusively to men | 11 (5.2) |
| **Sexual Behavior** | |
| Exclusively with women | 169 (79.7) |
| Predominantly with women only occasionally with men | 3 (1.4) |
| Equally with women and men | 1 (0.5) |
| Exclusively to men | 15 (7.1) |
| Neither women nor men | 24 (11.3) |

*2.2. Measures and Procedure*

Traits of pathological narcissism were evaluated with the pathological narcissism inventory (PNI) [11]. This 52-item scale uses a response format that ranges from 0 (=not at all like me) to 6 (=very much like me) and allows the assessment of both grandiose (GN) and vulnerable traits (VN) of pathological narcissism (Cronbach's α for GN: 0.86; α for VN: 0.93). Example items are "I often fantasize about being admired and respected" or "I help others in order to prove I'm a good person".

Sexual sex-esteem was assessed through the sexual self-esteem scale (SSE) [41]. The SSE includes 10 items rated from −2 (=disagree) to 2 (=agree) that assess the extent to which a person feels competent, confident, and skilled in their sexuality with a partner (i.e., I am a good sexual partner or I am not very confident in sexual encounters). The SSE showed good internal consistency (Cronbach's α = 0.93).

Sexual satisfaction was assessed through a single item from the sexual complaints screener for men (SCS-M) [42]. This is a comprehensive self-report tool addressing all areas of sexual functioning (sexual interest/desire, arousal, premature and delayed ejaculation, sexual pain, anxiety relating to penis size and shape, and sexual satisfaction). The SCS-M consists of 10 questions focusing on the last six months. The sexual satisfaction item ("During the last 6 months, my sexual life has been . . . ") is rated on a 5-point Likert scale (1 = very unsatisfying; 5 = very satisfying).

*2.3. Statistical Analyses*

All analyses were performed using IBM SPSS version 26.0 (IBM corp. 2019, Armonk, NY, USA). In a preliminary step, correlations between all variables were investigated. According to recent recommendations [43], we tested correlations of grandiose narcissism with sexual satisfaction and sexual self-esteem, controlling for vulnerable narcissism. In order to ascertain whether SSE might have a role in explaining the association between pathological narcissism and sexual satisfaction, we carried out two mediation models through PROCESS version 3.4.1 [44], one for grandiose narcissism and one for vulnerable narcissism. Again, in the model testing the effect of grandiose narcissism, we included vulnerable narcissism as a covariate [43]. Significance of indirect effects was established through bootstrapping procedures (i.e., 5000 bootstrapped samples): in particular, a 95% bootstrap confidence interval (CI) not containing zero indicates significant indirect effects [45]. Note that PROCESS also provides estimates of direct effects (i.e., unique effects of narcissism on sexual satisfaction, after controlling for the mediator) and total effects (i.e., the overall effect of narcissism on sexual satisfaction).

**3. Results**

Descriptive statistics and correlations are reported in Table 3. As shown, vulnerable narcissistic traits were negatively associated with both sexual satisfaction and sexual self-esteem. Grandiose narcissism showed to be positively correlated with sexual self-esteem, while the association of grandiose narcissism with sexual satisfaction was not significant.

The test of mediation for vulnerable narcissism showed that VN had a negative and significant total effect on sexual satisfaction (β = −0.15, *p* < 0.05). When considering the mediating effect of SSE (see Figure 1, left panel), the direct effect of vulnerable traits on sexual satisfaction was no longer significant, though its indirect effect through SSE was (β = −0.07, SE = 0.03; CI = (−0.14, −0.02)). In other words, the higher traits of vulnerable narcissism the lower sexual self-esteem, and the lower sexual self-esteem the lower sexual satisfaction in the last six months.

**Table 3.** Descriptive statistics and correlations.

| | SSE | Sex-S | VN | GN [1] |
|---|---|---|---|---|
| Sexual self-esteem (SSE) | - | 0.36 ** | −0.21 ** | 0.17 * |
| Sexual satisfaction (Sex-S) | - | - | −0.15 * | 0.09 |
| Vulnerable narcissism (VN) | - | - | - | - |
| Grandiose narcissism (GN) | - | - | - | - |
| M | 3.56 | 4.12 | 3.08 | - |
| SD | 0.84 | 1.29 | 0.68 | - |

[1] For grandiose narcissism, we computed partial correlations controlling for vulnerable narcissism. * *p* < 0.05; ** *p* < 0.01.

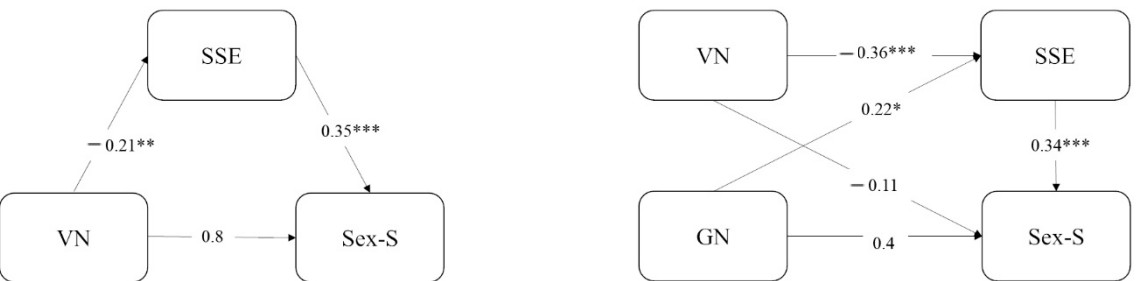

**Figure 1.** Mediation Model 1 (left panel) and Model 2 (right panel) predicting sexual satisfaction. Note: VN = vulnerable narcissism; GN = grandiose narcissism; SSE = sexual self-esteem; Sex-S = sexual satisfaction. Estimates are standardized. Goodness of fit for Model 1 (left): $R^2$ = 0.37, F(2, 208) = 16.65, *p* < 0.001. Goodness of fit for Model 2 (right): $R^2$ = 0.37, F(3, 207) = 11.16, *p* < 0.001. * *p* < 0.05; ** *p* < 0.01; *** *p* < 0.001.

The test of mediation for grandiose narcissism, on the other hand, showed that the total effect of grandiose narcissism on sexual satisfaction was not significant (β = 0.12, *p* = 0.17). Its direct effect on sexual satisfaction after including the mediator was also non-significant (see Figure 1, right panel). Nonetheless, the model showed that the indirect effect of grandiose narcissism through SSE was positive and significant (β = 0.08, SE = 0.03, CI = (0.01, 0.15)), suggesting mediation. In other words, the higher grandiose narcissism the higher sexual self-esteem and—in turn—the higher sexual satisfaction in the last six months.

## 4. Discussion

This study investigated the mediating role of sexual self-esteem in the association between manifestations of pathological narcissism and sexual satisfaction. Our results show that individuals high in vulnerable narcissism are prone to experiencing lower sexual satisfaction, while individuals high in grandiose narcissism are not. Our findings suggest that sexual self-esteem is a significant mediator both when considering grandiose and vulnerable narcissism. It is of note that several items of the SSE scale relate to the perception of one's sexual skills and performance (e.g., "I am better than most men at sex"), whereas the communal aspects of self-esteem are more neglected [46]. Hence, the sexual self-esteem score is not based on one's perception of his ability to be kind and loving in sex, nor to detect and fulfill the partner's needs. In this sense, our findings are interesting as they suggest that sexual satisfaction in those high in narcissism is highly dependent on one's

own perceived performance. Both our models even indicate full mediation, although this should be interpreted with caution because there is no guarantee that our models saturate all possible pathways leading from narcissism to sexual satisfaction [47]. Yet, results suggest that one's perceived ability in sexual performances represents a crucial mechanism contributing to sexual satisfaction in those high in grandiose narcissism, as well as to sexual dissatisfaction in those high in vulnerable traits. In this sense, our study mirrors and extends insights on the importance of self-reassuring motives in male narcissists' sexual life [30]. Furthermore, it is in line with the finding that a high or inflated sense of one's sexual skills is related to higher sexual satisfaction [6,31].

As for grandiose narcissism, it is interesting to further note that its significant indirect effect through SSE occurred in the context of a non-significant total effect of grandiose narcissism on sexual satisfaction. According to Baron and Kenny [48], the presence of a significant total effect would be a necessary condition to test mediation. Indeed, the relationship between the independent and the dependent variable (in our case, grandiose narcissism and sexual satisfaction) is traditionally tested prior to mediation to determine whether there is an effect to mediate at all: if not, it is assumed that no mediation can exist in the absence of an overall total effect. A full discussion of this issue is beyond the scope of our work. However, it is important to note that our findings are not unique as significant indirect effects can and do occur in the absence of a significant total effect [47]. Indeed, as Rucker and colleagues [47] argue, the total effect of a certain independent variable can be theoretically conceived as the sum of several direct and indirect effects (measured or unmeasured), not necessarily converging in the same direction. In this sense, opposing "lower-order" effects can suppress each other, obscuring a total effect at a less fine-grained level of analysis. For this reason, Rucker and colleagues suggest that overemphasizing the importance of a significant total effect in mediation analysis may be misleading.

As for our data, our findings indicate that, at least in a global sense, those high in grandiose narcissism are more satisfied with their sexual life and this is largely accounted for by the positive consideration of their personal abilities in sexual intercourse. At the same time, following the mentioned paper of Rucker et al. [47], we may hypothesize that PNI grandiose narcissism also includes unique features that do not promote sexual satisfaction or even exert a negative effect on it (directly or indirectly), resulting in a non-significant total effect of PNI grandiose narcissism. At this stage, our data only obliquely hint at this possibility. Whether, which, and why features of grandiose narcissism are uniquely unrelated or negatively related to reduced self-satisfaction in sexual life remains however an open question to examine in future research. We may speculate that grandiose fantasies, one of the components of PNI grandiose narcissism, may exert this opposing effect on sexual satisfaction, at least in part. Indeed, growing evidence suggests that this dimension of grandiose narcissism can be related to internalizing problems, as if producing "rebound" effects on one's self-perception [49,50].

*Clinical Implications*

Sexual satisfaction is not only an important element of individual well-being, it is also an indicator of relationship satisfaction. The present study shows that for individuals with high traits of pathological narcissism, sexual satisfaction is linked to sexual self-esteem. In other words, the representation of one's sexual skills with a partner are indicators of sexual satisfaction in individuals with pathological narcissistic traits. In particular, people with vulnerable traits seem to represent themselves as less skilled in sex with their partners. On the contrary, individuals with grandiose traits represent themselves as skilled lovers. This finding has important implications in working with individuals and couples in the context of psychotherapy or sexological counseling. Clinical work on sexuality with individuals with narcissistic traits should in fact be based primarily on the self-representation of one's own skills and abilities in sex. In the case of grandiose traits, the literature shows that abnormal self-representation can bring more distress to the partner than to the person [10]. The focus of the work could be to provide a more realistic perception of one's sexual

performance by opening a dialogue with the partner on their mutual needs, with an indication for couple therapy. In the case of individuals with higher vulnerable traits, the indication could be that of individual work on sexual self-esteem and self-perception of sexual performance. In fact, individuals with vulnerable traits report personal distress [10].

In general, personality traits can be an aspect to keep in mind in the clinical work with clients around sex and sexuality, including the self-representation as a more or less skilled lover.

## 5. Conclusions

The present study highlights how vulnerable narcissistic traits are negatively associated with sexual satisfaction. Sexual self-esteem fully mediates this association. On the contrary, grandiose narcissistic traits are not directly associated with sexual satisfaction, but with sexual self-esteem only, which explains the indirect effect of grandiose traits on sexual satisfaction. In conclusion, sexual self-esteem in personality configurations with high pathological narcissistic traits accounts for the relationship between narcissistic traits and sexual satisfaction.

*Limitations*

The present study is not without limitations. First, our participants represent an online convenience sample. This has had the effect of sampling men of limited sexual diversity with only a minority of our sample identifying as homosexual, bisexual, or other. Moreover, sexual satisfaction in the last six months was measured with a single item. This measure did not allow us to better nuance the different components of sexual satisfaction. Future studies may explore in more detail the relationship between narcissistic personality traits and satisfaction in diverse samples or focus only on individuals identifying as a sexual minority. Furthermore, the relationship between narcissistic traits and different components of sexual satisfaction should be explored in more depth.

**Author Contributions:** Conceptualization, A.P. and R.D.P.; methodology, A.P. and R.D.P.; formal analysis, A.A.; writing—original draft preparation, A.A. and M.D.S.; writing—review and editing A.A. and M.D.S.; supervision and revision, A.P. and R.D.P. All authors have read and agreed to the published version of the manuscript.

**Funding:** This research received no external funding.

**Institutional Review Board Statement:** The study was approved by the Ethics Committee of the University of Milano - Bicocca, with the ethical code num: 180, 2015.

**Informed Consent Statement:** Informed consent was obtained from all subjects involved in the study.

**Data Availability Statement:** The data presented in this study are available on request from the corresponding author. The data are not publicly available due to privacy reasons.

**Conflicts of Interest:** The authors declare no conflict of interest.

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
