# Peer review of "Narcissistic Personality Traits and Sexual Satisfaction in Men: The Role of Sexual Self-Esteem"

_sexes, doi:10.3390/sexes2010002_

Round 1

Reviewer 1 Report

The paper is of an interesting topic and well written in a condense way.  Argumentation is easy to follow.  Earlier research is presented in a balanced way.  The research method is appropriate for the research question and is well documented.  Conclusions are trustworthy deduced from the empirical results, and practical implications of the work are given.  I especially like the way the authors are careful in their conclusions, they are aware that their data is not telling everything and that other factors than studies might be active in the setting as well. I like this cautioness.

Author Response

We thank the reviewer for their comment on the paper.

Reviewer 2 Report

Nice study. The results are clear.

Please add a paragraph in the discussion section to address the limitations in the current study.

Author Response

Nice study. The results are clear.

Please add a paragraph in the discussion section to address the limitations in the current study.

We thank the reviewer for their comments. A paragraph of limitations was added.

Reviewer 3 Report

This is an interesting and well-written article in which the authors studied the role of sexual self-esteem in sexual satisfaction in men with narcissistic personality. Narcissistic personality was divided into grandiose and vulnerable since they show different patterns of associations regarding sexual outcomes; vulnerable narcissism was associated to lower self-esteem and lower sexual satisfaction, whereas grandiose narcissism was associated to higher self-esteem; in this case, sexual satisfaction seemed to be dependent on their own perception of sexual skills and performance.

In my opinion the introduction, methodology, results, and discussion are well presented; the clinical relevance of the study is well discussed in terms of both the relationship and the individual, which is important because pathological narcissism also affects the partner's sexual satisfaction, with implications in the relationship itself.

I just have some minor suggestions.

In the Methods, section 2.2, I suggest including all questionnaires used in this study as supplementary files to provide the reader a better understanding of the traits of pathological narcissism, sexual self-esteem, and sexual satisfaction. 

A minor observation, in Table 2, Sexual Attraction “Predominantly to women only accidentally to men” (instead of me)

Author Response

This is an interesting and well-written article in which the authors studied the role of sexual self-esteem in sexual satisfaction in men with narcissistic personality. Narcissistic personality was divided into grandiose and vulnerable since they show different patterns of associations regarding sexual outcomes; vulnerable narcissism was associated to lower self-esteem and lower sexual satisfaction, whereas grandiose narcissism was associated to higher self-esteem; in this case, sexual satisfaction seemed to be dependent on their own perception of sexual skills and performance.

In my opinion the introduction, methodology, results, and discussion are well presented; the clinical relevance of the study is well discussed in terms of both the relationship and the individual, which is important because pathological narcissism also affects the partner's sexual satisfaction, with implications in the relationship itself.

I just have some minor suggestions.

In the Methods, section 2.2, I suggest including all questionnaires used in this study as supplementary files to provide the reader a better understanding of the traits of pathological narcissism, sexual self-esteem, and sexual satisfaction.

We thank the reviewer for their comments. Unfortunately, we are unable to provide all the questions of the study as supplementary materials for copyright purposes, particularly regarding the measures of pathological narcissism. But we provided example of items for clarity.

A minor observation, in Table 2, Sexual Attraction “Predominantly to women only accidentally to men” (instead of me)

We corrected the typo.

Reviewer 4 Report

Sexes

November 11th, 2020

Subject: review of the article titled: “Narcissistic Personality Traits and Sexual Satisfaction in Men: The role of Sexual Self-Esteem” (ID: sexes-1024740)

Review Report

I really appreciated the opportunity to read this paper about narcissistic personality traits and sexual satisfaction in men. The study explores the relationship between narcissistic traits and sexual satisfaction in men, testing the mediation effect of sexual self‐esteem.

This is an interesting study which would be of interest to Sexes readers. Overall, I commend the authors for the relevancy of their work, which makes a significant contribution to the existing literature on the subject and is methodologically sound. The study aims are quite original and well defined. The conclusions are consistent with the arguments presented.

Very minor points:

Pag. 3: “(Wolff et al., 2017).” I cannot found Wolff et al in the reference section.

Instruments: I would suggest adding all the items used in the Wolff and colleagues’ (2017) measure. Furthermore, did the authors measure sex identity – How a person identifies physically: female, male, in between, beyond, or neither?

Table 2: I can see in table 2 that 11.3% of the participants declared their sexual behaviour as the following: “neither women nor men.” What does it mean? Are these participants without “sexual experiences”? If yes, how to measure sexual self-esteem (i.e., I am a good sexual partner or I am not very confident in sexual encounters).?

Line 118: Authors reported “sexual sex-esteem”. Correct with “sexual self-esteem”.

Limitation section: In spite of the findings stemming from the study, limitations should be reported. This preliminary study should be extended to a larger balanced sample. For instance, it would be helpful to include more sexual minority participants. Concerning this point, did the authors expect any relevant differences in the results obtained?

Author Response

I really appreciated the opportunity to read this paper about narcissistic personality traits and sexual satisfaction in men. The study explores the relationship between narcissistic traits and sexual satisfaction in men, testing the mediation effect of sexual self‐esteem.

This is an interesting study that would be of interest to Sexes readers. Overall, I commend the authors for the relevancy of their work, which makes a significant contribution to the existing literature on the subject and is methodologically sound. The study aims are quite original and well defined. The conclusions are consistent with the arguments presented.

Very minor points:

Pag. 3: “(Wolff et al., 2017).” I cannot found Wolff et al in the reference section.

Thank you for your comment. Reference was added.

Instruments: I would suggest adding all the items used in the Wolff and colleagues’ (2017) measure. Furthermore, did the authors measure sex identity – How a person identifies physically: female, male, in between, beyond, or neither?

The present work focused only on assigned male at birth individuals. The question was used as an inclusion criterion for the study.

Table 2: I can see in table 2 that 11.3% of the participants declared their sexual behaviour as the following: “neither women nor men.” What does it mean? Are these participants without “sexual experiences”? If yes, how to measure sexual self-esteem (i.e., I am a good sexual partner or I am not very confident in sexual encounters).?

Thank you for your comments. As clarified in the text, we asked about the participants’ sexual behavior in the last six months.

Line 118: Authors reported “sexual sex-esteem.” Correct with “sexual self-esteem.”

Thank you for your comment. We rephrased.

Limitation section: In spite of the findings stemming from the study, limitations should be reported. This preliminary study should be extended to a larger balanced sample. For instance, it would be helpful to include more sexual minority participants. Concerning this point, did the authors expect any relevant differences in the results obtained?

As suggested by other reviewers, we included a limitations section.